# Stent for life initiative in Portugal: Progress through years and Covid-19 impact

**Ernesto Pereira**[1,2,3]*, **Rita Calé**[3], **Ângela Maria Pereira**[4,5,6], **Hélder Pereira**[3,7], **Luís Dias Martins**[8]

**1** ISCTE-Instituto Universitário de Lisboa, Lisboa, Portugal, **2** Escola Superior de Saúde da Cruz Vermelha Portuguesa, Portuguesa, Portugal, **3** Cardiology Department, Hospital Garcia de Orta, EPE, Almada, Portugal, **4** Physiotherapy Department, Escola superior de Saúde Egas Moniz, Laranjeiro–Almada, Portugal, **5** Centro de Investigação Interdisciplinar Egas Moniz, Almada, Portugal, **6** Physiotherapy Department, Hospital Garcia de Orta, EPE, Almada, Portugal, **7** Faculdade de Medicina de Lisboa, Lisboa, Portugal, **8** BRU-ISCTE Business Research Unit, Lisboa, Portugal

* ejfapereira@gmail.com

## Abstract

### Background

During Stent for Life Initiative in Portugal lifetime, positive changes in ST elevation myocardial infarction treatment were observed, by the increase of Primary Angioplasty numbers and improvements in patients' behaviour towards myocardial infarction, with an increase in those who called 112 and the lower proportion attending non primary percutaneous coronary intervention centres. Despite public awareness campaigns and system educational programmes, patient and system delay did not change significantly over this period. The aim of this study was to address the public awareness campaign effectiveness on peoples' behaviour facing STEMI, and how Covid-19 has affected STEMI treatment.

### Methods

Data from 1381 STEMI patients were collected during a one-month period each year, from 2011 to 2016, and during one and a half month, matching first lockdown in Portugal 2020. Four groups were constituted: Group A (2011); Group B (2012&2013); Group C (2015&2016) and group D (2020).

### Results

The proportion of patients who called 112, increased significantly (35.2% Group A; 38.7% Group B; 44.0% Group C and 49.6% Group D, p = 0.005); significant reduction was observed in the proportion of patients who attended healthcare centres without PPCI (54.5% group A; 47.6% Group B; 43.2% Group C and 40.9% Group D, p = 0.016), but there were no differences on groups comparison. Total ischemic time, measured from symptoms onset to reperfusion increased progressively from group A [250.0 (178.0–430.0)] to D [296.0 (201.0–457.5.8)] p = 0.012, with statistically significant difference between group C and D (p = 0.034).

**Data Availability Statement:** Data cannot be shared publicly because it belongs to the National Cardiology Data Center (CNCDC) from the Portuguese Society of Cardiology. According to its regulations, only investigators from the

participating centre (any portuguese cardiology centre) have access to the aggregated data upon permission. Data are available from the CNCDC's Registries organising committees (contact via cncdc@spc.pt) for researchers who meet the criteria for access to confidential data. Other researchers who are not part of a participating center must submit a written permission request explaining the reasons why they need the data. CNCDC statement uploaded as "Third-party source information availability".

**Funding:** The author(s) received no specific funding for this work.

**Competing interests:** The authors have declared that no competing interests exist.

## Conclusions

During the term of SFL initiative in Portugal, patients resorted less to primary health centres and called more to 112. These results can be attributed the public awareness campaign. Nevertheless, patient and system delays did not significantly change over this period, mainly in late years of SFL, probably for low efficacy of campaigns and in 2020 due to Covid-19 pandemic.

## Introduction

The Stent for Life (SFL) programme which was effective from 2008 to 2016, in 23 countries, mainly in Europe, intended to improve significantly the patient access to the lifesaving primary percutaneous coronary intervention (PPCI), in order to reduce mortality and morbidity in patients suffering from an acute myocardial infarction (AMI), also known as ST elevation myocardial infarction (STEMI) [1,2]. In Portugal this programme was active between 2011 and 2016 [2]. The denomination changed in 2016 to Stent Save a Life (SSL), which had as primary intention to extend this mission globally, following the increasing needs and adapting it to the specific demands of each world region. The principles are the same as in Stent for Life, to improve the delivery of care and patient access to PPCI, reducing mortality and morbidity in patients with acute coronary syndromes (ACS). In addition to increasing the number of PPCI per million inhabitants, other objective of SFL and SSL is to reduce reperfusion times, which is crucial given the total ischaemic time, (time from symptom onset to reperfusion therapy), has prognostic impact [3–5].

The Portuguese SFL initiative, since it was implemented, carried out a public awareness campaign to raise public consciousness of myocardial infarction's (MI) symptoms and to persuade people to call 112 (European Emergency Medical Services–EMS number) immediately. This campaign was developed under the international SFL slogan 'Act now. Save a life'. The first results pointed that its main impact was on patient delay with great improvement observed when compared the baseline results to those obtained one year later [6]. However, this improvement was attenuated through the years and five years after implementation of SFL, time to treatment and its components, patient and system delay came to values near to those observed at baseline in 2011. Nevertheless, other indicators, such as the percentage of patients who called the EMS number, went to healthcare centres without PPCI facilities by their own means or were primarily and/or secondarily transported by EMS, could be more sensitive ways to assess whether the SFL initiative led to positive changes [2].

In 2013, we carried out a market research study in collaboration with ISCTE—Instituto Universitário de Lisboa, to assess the result of the campaign and the knowledge of the population about ST elevation myocardial infarction (STEMI) and what is the correct action after the onset of symptoms. In 2016 another market research study, was ordered to a well-known Portuguese agency of market research, Pitagorica, with the same purpose. Both studies were unpublished, only the first was presented in several cardiology meetings, and press releases of results were made. These studies may help to understand the reason for the draw-back in patient delay.

In December 2019, a new coronavirus (SARS-CoV-2) was identified in Wuhan (China) caused COVID-19 [7]. In 2020, COVID-19 spread rapidly worldwide and was officially recognized as a global pandemic by the World Health Organization (WHO) on March 11 [8]. Recent statistical data on the pandemic show that more than 659 million people worldwide

were infected, and the number of deaths had reached 6.679 million, and still growing. In Portugal these numbers reach 5.5 million of people infected and 25 thousand deaths online December 22; 16:30 min [9].

As it is a highly contagious disease, with no effective vaccine available at the beginning, a large part of human activities had to be prohibited all over the world to contain its proliferation. On 2020 March 18th, the Portuguese government declared the first State of Emergency and decreed prohibitions to all citizens which included exits to public roads, lockdowns in commerce, companies, schools, universities, tourism, and restrictions in the movement of people among regions and countries [10].

The COVID-19 caused considerable and multifactorial stress on health care systems worldwide. The most evident issue was related to the treatment of the respiratory syndrome directly caused by the SARS-CoV-2 infection. During the most difficult phase of the *first wave*, a considerable number of Portuguese hospitals had to expand their intensive care capacity exponentially and adapt a large part of their overall hospital capacity to treat patients with COVID-19. This adaptation on national healthcare service compromised the accessibility of diagnosis and treatment in major health areas, with short- and long-term consequences that are not yet fully understood. Elective admissions, diagnostic tests, and interventions considered not urgent were cancelled to allocate resources to the huge increase in infected patients admission. Nevertheless most of health-care programmes for urgent conditions such as AMI and stroke continued active during this period, but the numbers of patients attending hospitals for these conditions fell, mainly in the first weeks of the lockdown. The general belief was that patients experiencing symptoms compatible with AMI were not attending hospitals due to fear of being exposed to an environment with high risk of SARS-CoV-2 infection. Reports all over the world have suggested a decrease in the number of patients presenting to hospitals with STEMI [11].

The Covid-19 pandemic brought new challenges to countries health systems in treatment of infected patients and as well as on the way to prevent its spread. Also, as we said before, some reports highlighted a decrease in the number of STEMI patients attending hospitals in Europe and North America [12–14]. However, we have limited information on how this decrease affected STEMI networks in terms of delays to reperfusion [14]. On this topic, particularly, patient delay may be affected by patient awareness to identify symptoms and act properly. As said before, the Portuguese SFL initiative, since it was implemented, carried out a public awareness campaign to raise public awareness of the MI's symptoms and to encourage people to call 112 immediately [6]. Nevertheless, patient delay was, in 2016, already as long as in 2010 [2].

The objectives of this study were to address the effectiveness of public awareness campaign on peoples' behaviour facing STEMI and how Covid-19 has impacted STEMI treatment in the Portuguese Healthcare.

## Methods

In 2011, the SFL task force identified three main vectors to design an action plan with specific interventions: 1) the population: developing an awareness campaign to teach how to recognise infarction symptoms, to call 112 as quick as possible and not to go to the hospital by their own means, 2) EMS: establishing that EMS directly contact centres with PPCI units, sent ECG to these units in advance and get EMS collaboration to carry out the secondary transport between the hospitals without and with PPCI unit. Additionally, it was initiated an educational program in a ACS targeting EMS healthcare professionals, denominated STEMINEM. 3) hospitals: Besides the meetings aiming to simplify the implementation of a national network of

PPCI 24/7, it was initiated the educational program in ACS, *Stent network Meeting* which targets emergency department healthcare professionals, denominated STEMICARE, extended to all national mainland.

To evaluate the success of these actions, specific assessment methods were established: qualitative assessment of *patient and system delay–Study Moments*, as well as market research assessment studies.

Two market researches studies, were made to assess population knowledge in early and late years of SFL activities in Portugal. The first was carried out in 2012, by a partnership between SLF and ISCTE, with the aim to know the recall level of the Act Now Save a Life campaign developed by SFL, and to access the awareness level of Portuguese population regarding MI symptoms and the need to dial the national emergency number 112. The methodology used was a phone interview, applying a structured questionnaire with open and closed questions, directed to 1000 subjects, living in Portugal, ≥ 18 years old, both gender selected by quota sampling, based on NUTS II and age.

The second was carried out in 2017, by a well-known Portuguese market research agency–Pitagorica–ordered by Stent Save a Life Portugal with the name *The Portuguese people and Myocardial Infarction*. This study had a broader scope concerning Portuguese healthcare usage, but mainly to evaluate population knowledge on MI, and its perception on MI severity and associated risk factors. The research was applied to Portuguese population over 15 years old, by telephone interview. The sample was collected in November 2017 and consisted of 1044 subjects, (confidence level of 95.5%, and margin of error of 3.09%).

The *Study Moments* were prospective transversal studies during one-month per year (2011–2016), conducted in all hospitals with a PPCI programme in mainland Portugal and integrated in the STEMI fast track (the national STEMI network), to assess the campaign effect over the years for the duration of SFL initiative. The same methodology, "Study Moments" named by us "Moment Covid", gathered in 2020, was used to assess data from the first months of Covid-19 in Portugal, March to May.

Data from 1381 patients suspected of suffering STEMI with less than 12 hours of evolution and referred to PPCI, admitted in 18 main land Portuguese cardiology centres, were collected during a one-month period each year, from 2011 to 2016, and during one and a half month, matching with the first lockdown in Portugal 2020. All enrolled patients data were collected from the Cardiovascular Intervention National Registry, managed by the Portuguese Cardiology Data Collection Centre (CNCDC), which is a department of the Portuguese Society of Cardiology (PSC), authorized to gather data on patients for research proposals, that complies with the Regulation (EU) 2016/679 from the General Data Protection Regulation, regarding the processing of personal data and on the free movement of such data (http://www.spc.pt/CNCDC) [15,16]. All enrolled patients gave their written informed consent for P-PCI and for data collection by the CNCDC. Authors did not have access to information that could identify individual participants during or after data collection. This manuscript adheres to the authors' national ethics guidelines which reflect the Declaration of Helsinki.

Four groups were constituted: Group A (patients included in Moment 0–2011); Group B (patients included in Moments 1 and 2–2012&2013); Group C (patients included in Moments 4 and 5–2015&2016) and group D (patients included in Moment Covid– 2020).

This study aims to compare the results of Group A with Group B, to evaluate the effect of campaign in the early period of Portuguese SFL; Group B with Group C, to evaluate the effect of a long period campaign; and Group C with Group D to evaluate the Covid-19 effect.

Time assessment was done according the methodology used previously in Portuguese SFL studies [2,6]. The first medical contact (FMC) was defined as the time of arrival of medical and/or paramedical staff to assist the patient for prehospital care or the time of arrival at a

hospital. *Patient delay* was defined as the time between symptom onset and FMC, it was considered a continuous variable and expressed in minutes. *System delay* was defined as the time from FMC to the beginning of reperfusion therapy, either with balloon, wiring or mechanical thrombectomy, and it was considered a continuous or categorical variable (cut-off value = 90 minutes).

*Door-to-balloon* time (D2B) was defined as the time from admission to a PPCI capable hospital to the reperfusion therapy and it was evaluated as a continuous or categorical variable (cut-off value = 60 minutes). Total ischemic time was defined as the time between the symptom onset and reperfusion and was evaluated as a continuous or categorical variable (cut-off value = 120 minutes).

## Statistical analysis

Continuous variables were presented as mean standard deviation (SD), the normality of data was assessed by Kolmogorov-Smirnov test, and the equality of variances was assessed by Levene test. T-student test was used to compare the mean of variables presenting normal distribution or equality of variances (e.g., age). Categorical variables were presented as percentage and the proportions were compared by chi-square test or Fisher's exact test. Continuous variables that were not normally distributed, including patient delay and system delay, were presented as median and interquartile range (IQR), compared with Mann-Whitney U test for two independent samples, and with Kruskal-Wallis for more than two independent samples. Two-sided p-values <0.05, were considered statistically significant. All statistical analysis was performed using SPSS software for Windows version 21.

## Results

In the years 2011, 2012/2013, 2015/2016 and 2020, 1380 patients, from 18 national interventional cardiology centres, were enrolled. The demographic data and medical history of these patients are presented in Table 1. No statistically significant differences were observed concerning demographic data or previous history of coronary disease.

Time to treatment variables are expressed on Table 2. Total ischemic time, measured from symptoms onset to reperfusion increased progressively from group A [250.0 (178.0–430.0)] to D [296.0 (201.0–457.5.8)] p = 0.012, with statistically significant difference between group C and D p = 0.034.

Nevertheless, total delay within the limits set by the ESC and the American College Cardiology/American Heart Association guidelines, was achieved only by a small proportion of patients varying from 8.2% in Group A, with a little increase to B (10.0%) decreasing to 4,0%

**Table 1. Demographics and clinical background.**

| | (Group A) (N = 187) | (Group B) (N = 410) | (Group C) (N = 487) | (Group D) (N = 297) | p-value | p-value A vs B | p-value B vs C | p-value C vs D |
|---|---|---|---|---|---|---|---|---|
| Female gender, n (%) | 41 (21.9%) | 93 (23.3%) | 124 (25.9%) | 72 (24.8%) | 0.677 | 0.710 | 0.368 | 0.731 |
| Age (years), mean (SD) | 62.1 (13.7) | 62.1 (13.4) | 63.5 (12.9) | 63.5 (13.7) | 0.275 | 0.987 | 0.109 | 0.798 |
| Age ≥ 75 years, n (%) | 34 (18.4%) | 78 (19.5%) | 106 (22.4%) | 65 (22.7%) | 0.498 | 0.748 | 0.294 | 0.919 |
| Medical history of PCI, n (%) | 18(9.9%) | 44 (10.9%) | 56 (11.8%) | 31 (10.9%) | 0.918 | 0.713 | 0.668 | 0.707 |
| Medical history of CABG, n (%) | 1 (0.6%) | 3 (0.7%) | 9 (1.9%) | 4 (1.4%) | 0.430 | 0.802 | 0.141 | 0.640 |
| Medical history of MI, n (%) | 17 (9.7%) | 49 (12.2%) | 49 (10.4%) | 32 (11.3%) | 0.789 | 0.396 | 0.406 | 0.691 |
| Medical history of DM, n (%) | 28 (16.2%) | 87 (21.6%) | 122 (25.8%) | 55 (20.3%) | 0.047 | 0.137 | 0.141 | 0.087 |

PCI: Percutaneous coronary intervention; CABG: Coronary Artery By-pas Graft; MI: Myocardial; DM: Diabetes Mellitus.

**Table 2. Time to treatment variables.**

| Time (min) Med (Q25-Q75) | (Group A) (N = 187) | (Group B) (N = 410) | (Group C) (N = 487) | (Group D) (N = 297) | p-value | p-value A vs B | p-value B vs C | p-value C vs D |
|---|---|---|---|---|---|---|---|---|
| Time from symptom onset to FMC—patient-delay | 113.5 (63.5–211.5) | 94.5 (49.8–199.3) | 110.0 (55.8–210.0) | 120.0 (60.0–240.0) | 0.061 | **0,024** | 0.237 | 0.225 |
| Time from FMC to ECG | 16.0 (8.0–39.0) | 13.0 (6.0–30.0) | 14.0 (5.0–32.0) | 15.0 (7.0–32.0) | 0.222 | **0.046** | 0.633 | 0.372 |
| Time from general emergency service without PPCI to admission to PPCI hospital | 110.0 (85.0–164.0) | 117.0 (88.0–210.0) | 126.0 (93.8–195.0) | 144.5 (100.0–225.0) | 0.098 | 0.385 | 0.525 | 0.176 |
| Time from admission to PPCI hospital to reperfusion door-to-balloon time | 54.0 (30.0–99.5) | 54.5 (30.0–85.0) | 57.5 (30.0–96.0) | 63.0 (32.0–108.0) | 0.346 | 0.904 | 0.669 | 0.189 |
| Time from admission to PPCI hospital to CathLab | | 17.0 (10.0–63.0) | 25.0 (5.0–70.0) | 30.5 (2.0–81.0) | 0.108 | NA | **0.018** | 0.624 |
| Time from FMC to reperfusion—system delay | 115.0 (79.0–179.5) | 120.0 (89.0–187.5) | 130.0 (94.5–197.5) | 150.5 (109.0–231.3) | <0.001 | 0.410 | 0.098 | **0.005** |
| Time from symptom onset to reperfusion—total ischemic time | 250.0 (178.0–430.0) | 254.0 (165.0–386.8) | 269.0 (181.0–409.0) | 296.0 (201.0–457.5) | **0.012** | 0.332 | 0.154 | **0.034** |

FMC: First medical contact; EMG: Electrocardiogram; PPCI: Primary Percutaneous Coronary Intervention; CathLab: Catheterization laboratories.

on group D, without statistically significant differences in groups comparison (Fig 1)
p = 0.024, with a decrease in median time, which was 113.5 (63.5–211.5) in group A and 94.5 (49.8–199.3) in group B.

When we look for patient delay, we found significant differences only between groups A&B. Although there were no statistically significant differences, an increase of patient delay was observed from group B to C and from C to D, achieving on D a value similar to the one observed on group A.

When we look for patient delay, we found significant differences only between groups A&B. Although there were no statistically significant differences, an increase of patient delay was observed from group B to C and from C to D, achieving on D a value similar to the one observed on group A.

The proportion of patients who called 112, increased significantly (35.2% in group A; 38.7% in Group B; 44.0% in Group C and 49.6% in Group D, p = 0.005) (Fig 2). Despite the continuous increase all over the years, there were no significant differences when groups were

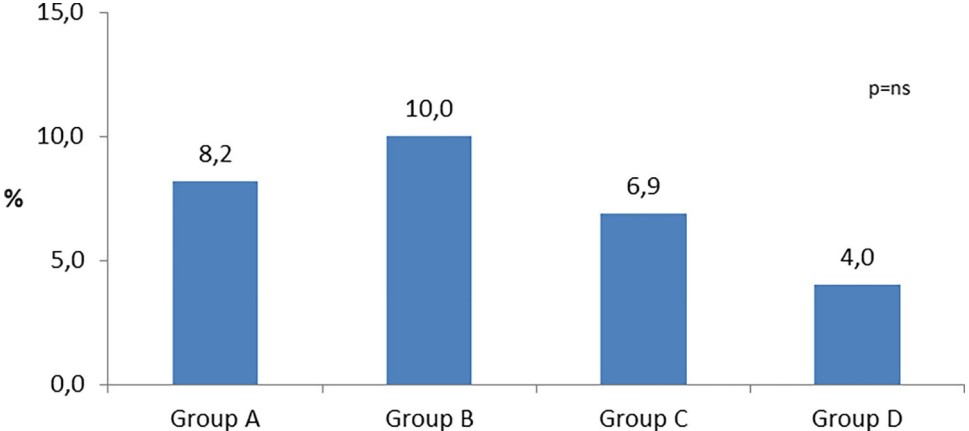

**Fig 1. Proportion of patients who met recommendations of the European Society of cardiology for total ischemic time < 120 minutes.**

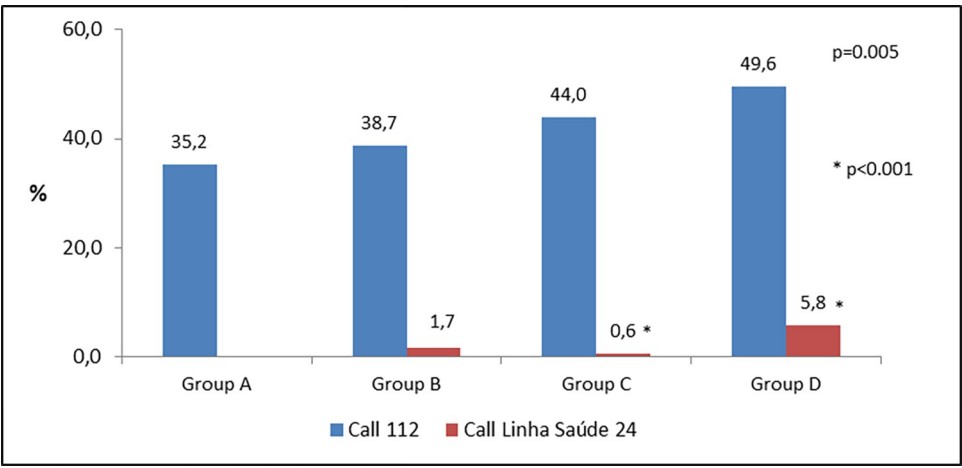

**Fig 2. Proportion of patients call for help by calling 112 and *Linha Saúde 24*.**

compared (A&B; B&C and C&D). Something similar occurred to the proportion of patients who called *Linha Saúde 24*, an official health call centre which advises patients about what to do when they have health issues.

This variable was not registered in Moment 0, so it was not possible to compare groups A&B, significant difference was observed between groups C&D p<0.001 (Fig 2). On the other hand, significant reduction was observed in the proportion of patients who attended health-care centres without PPCI (54.5% in group A; 47.6% in Group B; 43.2% in Group C and 40.9% in Group D, p = 0.016), but there were no differences on groups comparison. We found significant differences, as well, on those patients who attended primary care centres (p<0.001) with significant differences between groups A&B (20.3% - 9.7% p<0.001), B&C (9.7% - 5.1% p<0,009). Between C&D the difference was not statistically significant, but showed the same tendency (5.1% - 2.4% p<0.07) (Fig 3).

System delay was assessed by measuring FMC to treatment and its subdivisions, FMC to ECG (time to diagnosis); time from admission to PPCI centre to reperfusion (door to balloon).

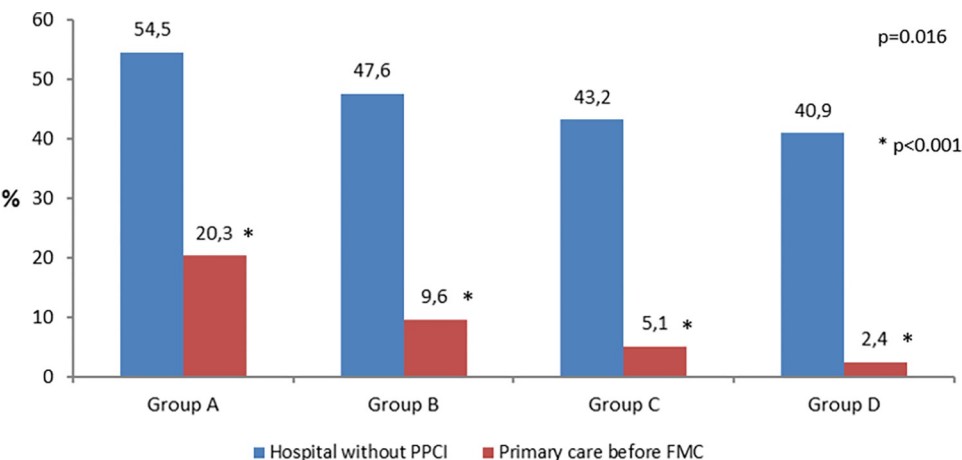

**Fig 3. Proportion of patients who attended healthcare facilities without possibility to perform primary percutaneous coronary intervention; and proportion of patients who attended primary care before First Medical Contact.** (PPCI) Primary Percutaneous Coronary Intervention; (FMC) First Medical Contact.

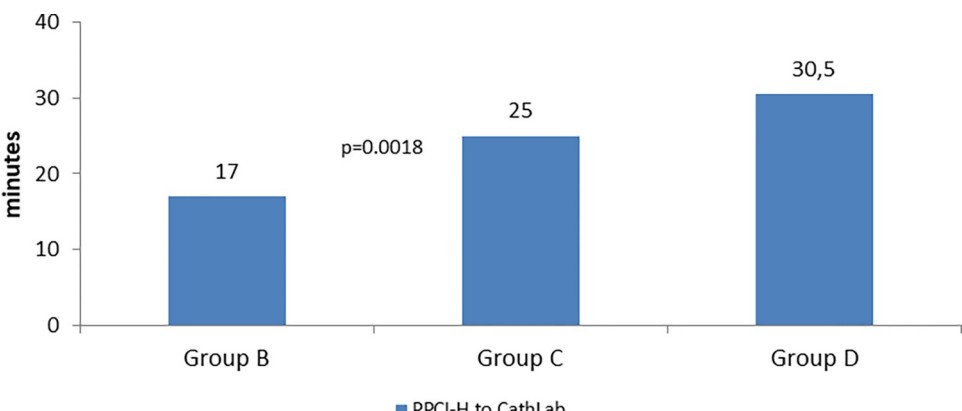

**Fig 4. Time in minutes from arrival to primary percutaneous coronary intervention hospital (PPCI-H) to Catheterization laboratories (CathLab).**

Total system delay presented significant differences ($p < 0.001$) for the overall periods, and only between groups C&D p = 0.05, with an increase on median time, which was 115.0 (79–179.5) min, on group A; 120.0 (89.0–187.5) min, on group B; 130.0 (94.5–197.5) min on group C and 150.5 (109.0–231.3) on group D. Although there were no statistically significant differences, patients who seek for help at a general emergency service without PPCI presented a slight increase on time from general hospital to admission in PPCI hospital for all groups. The same occurred with D2B time, which has remained practically unchanged for all groups. Time from admission in PPCI hospital to CathLab, was also measured to Groups B, C and D. This variable was introduced to assess how long patients take from arrival to PPCI hospital to enter in CathLab. It was registered an increase trough the periods, with statistically significant differences between group B to C (Fig 4).

Time from FMC to ECG presented an improvement comparing A to B groups reflected by a decrease of 3 min in median time (16 to 13 min, p = 0.046). From B to C, and from C to D there was a worsening, reaching a similar time between A (16 min) and D (15min) groups (Fig 5).

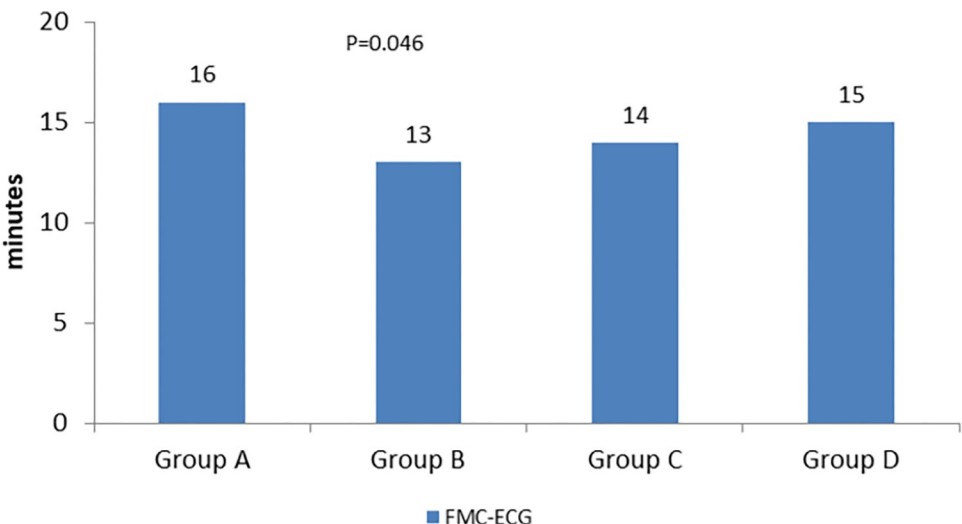

**Fig 5. Time to diagnosis confirmation measured in minutes. FMC-ECG** FMC–ECG: Time from first medical contact to electrocardiogram.

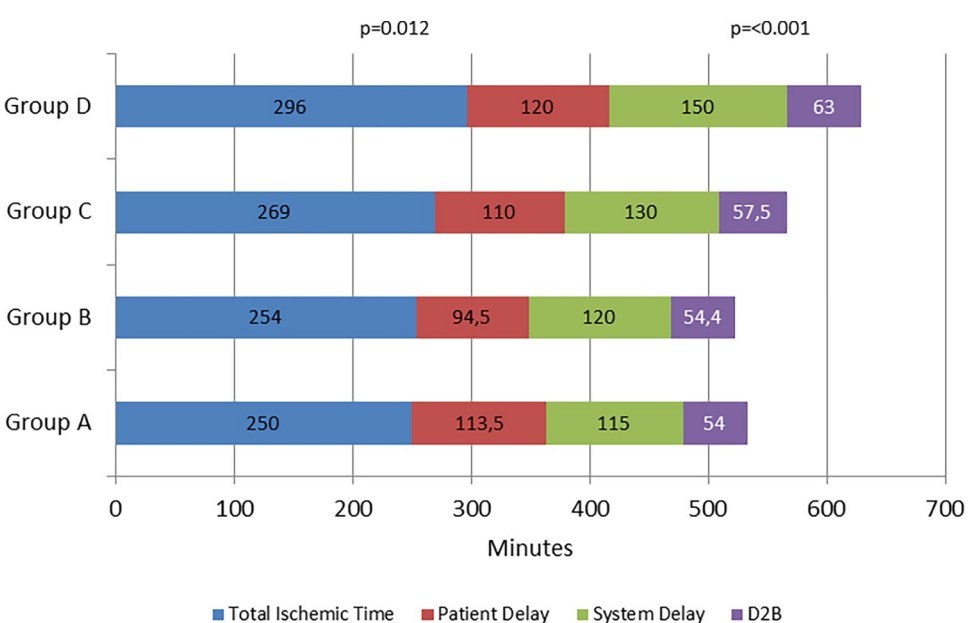

**Fig 6. Time to treatment and its subdivisions in minutes.** Total ischemic time: Time from onset symptoms to revascularization; Patient Delay: Time from onset symptoms to first medical contact; System Delay: Time from first medical contact to revascularization.

Total ischemic time, Patient Delay, System Delay and Door-Balloon are demonstrated in Fig 6.

For system delay components, proportion of patients who were within the limits set by the guidelines, for FMC to ECG ($\leq$ 10 min) was 32.7% on group A, with a significant increase to 43% on group B (p = 0.025), and remained constant in groups C and D (40% for both groups) Door-to-balloon ($\leq$ 60 min) remained nearly unchanged for all groups. Regarding global FMC to balloon time, system delay ($\leq$ 90 min) it decreased through the period with significant differences globally (p<0.001) it was 31.6% on group A, 27.2% on group B, 22.0% on group C and 13.9% on group D with significant differences, between groups C and D. For total ischemic time ($\leq$ 120 min) there was significant differences (p = 0.05) and it was achieved only by a small proportion of patients, shifting from 8.2% in Group A, with a little increase to B (10.0%), decreasing to 4,0% on groups C and D, without statistically significant differences between groups. Fig 7 represents the proportion of patients who were within the guideline's limits.

## Discussion

The Portuguese Association of Cardiovascular Intervention joined the SFL initiative in February 2011. This initiative was launched in 2008 by the European Association for Percutaneous Cardiovascular Interventions and EuroPCR and national cardiology societies of the participating countries [17]. The purpose of this initiative was to reduce mortality and morbidity in patients who suffer a STEMI in European countries, by improving patient access to PPCI, which is the recommended procedure to treat this kind of patients [18].

Before SFL, and despite having an acceptable coverage of PPCI centres and road networks, Portugal, had one of the lowest rates of PPCI implementation in Europe [6]. SFL objectives were: (i) to increase the use of PPCI to more than 70% of all STEMI patients; (ii) to achieve PPCI rates of more than 600 per million inhabitants per year; and (iii) to offer 24-hour 7 days

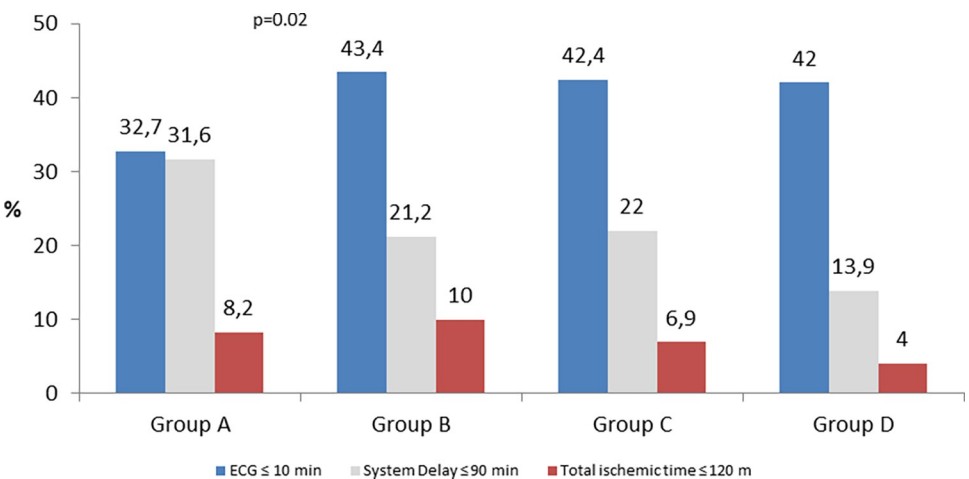

**Fig 7. Proportion of patients who met recommendations of the European Society of cardiology** First medical contact to electrocardiogram (ECG ≤10 min); First medical contact to reperfusion (system delay ≤ 90 min; time from onset symptoms to reperfusion.

a week (24/7) PPCI service at all invasive facilities in order to cover the countries' STEMI population needs.

The best benefits of reperfusion therapy in STEMI are achieved when this therapy is performed as soon as possible after the symptom onset, preferentially within the first 6 hours [18]. Therefore, its main objective was to increase timely access to PPCI, in order to reduce mortality and morbidity in patients with STEMI.

The SFL task force identified some barriers which needed interventions. First, patients' knowledge about recognising myocardial infarction symptoms was not adequate which made them ask for help too late. Also, patients didn't act correctly toward a MI, mostly they went to the hospital by their own means, instead of calling 112. Before SFL only 33% of the patients used the EMS number, 112, to call for assistance [6].

Second, the EMS and hospital networks performance presented long system delays [6]. In order to overcome these difficulties, several actions were designed and implemented concerning people's awareness to MI, and some measures targeted health professionals and to the national health system to increase their performance.

The public *Act now. Save a life* campaign was launched, with the help of a communication agency. Informative materials, including flyers, posters and videos, were produced to support the campaign. Public figures were invited to be ambassadors and partnerships were set up with the Portuguese Pharmaceutical Society, major national companies in different business areas (energy, entertainment, supermarkets, etc.) and local government bodies. During the five-year period, the campaign was ongoing in the press, on radio and television.

In 2013 the first market research, was conducted in together with ISCTE. This research was carried out on 1000 subjects, representing the Portuguese population and its distribution. Its main objectives were to know the recall level of the Act Now Save a Life Campaign developed by SFL to know the awareness level of Portuguese population regarding MI's symptoms and the need to dial the EMS number 112.

The majority of the responders (95%) knew the EMS number and 91% answered that they would use this number to call an ambulance in case of AMI. This finding may justify the improvement observed when comparing groups A and B, where an higher number of patients who call 112 was observed. However, only 24% of the responders were aware of AMI

symptoms, and 64% of the subjects thought that MI is a typical male disease, demonstrating that some campaign messages were not effective.

Other findings which may be related with campaign effectiveness was the decrease in the proportion of patients who attended healthcare centres without PPCI, as well as patients who attended primary care centres, leading to an improvement on patient delay, which decreased from a median of 113.5 (A) to 94.5(B). It is known that the sooner reperfusion therapy in STEMI is attained the better is the patient prognosis [3–5]. Shortening patient delay allows to decrease time to diagnosis and consequently to perform treatment earlier. The predictors which contribute the most for longer patient delay in Portugal were identified previously, by Pereira et al. [19], which are: age ≥75 years old, symptom onset between 0:00 AM and 8:00 AM, and primary care unit before FMC, on the other hand call 112-EMS and EMS transport to the PPCI facility was associated to shorter patient delay. Based on this information, it is possible to plan more efficient mass media campaigns, focused on minimizing the impact of these factors and targeting the specific groups (e.g. older patients). These actions should be important to reduce the patient delay, and allow timely treatment of STEMI patients.

In 2017 a Pitagorica market research study ordered by Stent Save a Life Portugal with the name The Portuguese people and Myocardial Infarction was applied to sample of 1044 subjects representative of Portuguese population over 15 years old, by telephone interview. This study assessed population knowledge on MI, and its perception on MI severity and associated risk factors. The results were unpublished, but in a press release made in 2018, it was written that people's knowledge about MI increased. Of the 1044 respondents, 95% associate chest pain with MI and the vast majority (96%) were aware that this pathology is an extremely serious disease that needs immediate treatment. Demonstrating the better knowledge about this pathology is the fact that 68% of the respondents answered that MI is diagnosed through an electrocardiogram, and 42% answered that the infarction results from the occlusion of a coronary artery. When asked about what they would do in the presence of a sign or symptom of Acute Myocardial Infarction, more than half of respondents (57%) stated that calling 112 would be the first option, although when asked about what they would do when faced with the symptoms of chest pain, with sweating, nausea and vomiting, only 38% reported calling 112. Given the symptoms presented, going to a hospital emergency would be the common practice for 27% of respondents in this study. Pereira et al [2] referred that this findings were in line with the positive results observed in the behaviour of the Portuguese people facing MI symptoms during the period in which the SFL initiative was in effect. Between 2011 and 2016, there was an increase in patients who called 112 (35.2% vs 46.6%) and a decrease in the percentage of patients who went by their own means to hospitals without interventional cardiology (54 .5% vs 42.4%). We found similar results, with an increased proportion of patients calling 112 (group A 35.2% vs group D 49.6%). A decrease on proportions of patients who attended healthcare centres without PPCI (54.5% in group A vs group D 40,9%), and of those patients who attended primary care centres (group A 20.3%; group B 9.7%; group C 5.1%; and group D 2.4%).

Nevertheless, when we compared patient delay between groups B&C, despite apparent improvement in patients' knowledge reflected by the increase calling to 112, the decrease in attendance recurrence to healthcare centres without PPCI, and to primary care centres, an increase was observed on patient delay. This may be related to lack of knowledge of MI symptoms, which led people to seek help later. It is well known if it is intended to change people's behaviour, it is necessary to change what they think about a certain topic [20,21]. For campaigns to produce behavioural changes and better results, they must be long lasting, their message must be reinforced, monitored and adjusted. In the other hand, another problem which may explain patient delay over the first five years may be related with an ad fatigue effect of the

campaign. This effect has been described before and is associated to campaigns which last too long on time without changing the message, and also when the message is repeated too much times. So, ad fatigue occurs when your audience sees your ads so often that they become bored with them and stop paying attention, reducing its effectiveness [22]. In 2016, SFL finished its programme and a new initiative was launched, SSL, sharing the same mission, principles and objectives of SFL, with the intention extend them worldwide. The new direction of SSL Portugal made changed the campaign name to *Each Second Counts* and reduced some public actions, maintained active the site https://www.stentsavealife.pt/, as the main vehicle of message diffusion. As not all the population accede, the campaign effectiveness is probably decreasing through the recent years.

In addition to the reduction of the campaign effect, patient delay increased even more in group D, which may certainly be related with Covid-19 pandemic. Several studies worldwide [11,14,23] report a significant decrease in patients admitted to hospitals with STEMI and consequently decrease in PPCI. We found this effect in *Moment Covid* survey, the patients admitted on first three weeks after the lockdown, March 18 to April 8, represented only 25% of the patients admitted between March 18 and May 6.

The decrease on STEMI admissions may be explained by the fear of becoming infected, which led people to avoid going to the hospital. An Italian study compared out-of-hospital cardiac arrest (OHCA) during pandemic period with same period of 2019, and referred an increase of 58% in OHCA, which supports the hypothesis of patients being avoidant of hospital attendance [24].

As it is known the best benefits of reperfusion therapy in STEMI are achieved when this therapy is performed, preferentially within the first 2 or 3 hours after the symptom onset [18].

Total ischemic time is the sum of patient delay, calculated from symptom onset to FMC and system delay, from FMC to PPCI. A total ischemic time longer than four hours has been identified as an independent indicator of mortality within a year of STEMI [25,26].

In our study total ischemic time had been increasing progressively from group A to D, with major difference between groups C and D. The same was observed in system delay progression. Pereira et al explained the increase in system delay, which reflected in total ischemic delay, as a consequence of widening access to less populated and more remote regions. The time of the system started earlier and is necessarily longer due to an increasing proportion of patients in remote regions calling the EMS number and receiving a prompt response. Thereby, improvements in prehospital management and patient transport may contribute to worsen system delay, because FMC contact was earlier and the counting to the system side starts at this point, even though it may also mean a substantial improvement in the system itself [2].

In general, delay to treatment in 2020 (Group D) has increased, significantly or not, in all partial times, patient delay, system delay, even in door-to-balloon, which was very stable all over the years. Covid-19 is the reason for this finding. At this period, despite an increase in patients' education about MI recognition and how to proceed if MI occurs, there was a feeling of fear to get infected with a new disease, barely known, with high mortality rate, which lead patients to contact medical help later. The increased calling to *Linha Saúde 24* means that patients seek for a diagnosis confirmation before getting the right help. The patient delay in group D increased 10 min in median, comparing with group C. On system side, indicators were even worse, system delay increased 20.5 min in median, D2B increased 6.5 min and time from admission to CathLab increased 5 min. Total ischemic increase 29 min in median. Additionally indicators of quality as the proportion of patients who were within the guidelines limits decreased, with exception of *Time from FMC to ECG* $\leq$ 10 min, which registered similar proportions. D2B $\leq$ 60 min decreased 5.4%, system delay $\leq$ 90 min decreased 8.1% and total ischemic time $\leq$ 120 min, decreased 2.9%. These findings are compatible with the constrains

that heath-services were submitted to, such as shorten of intensive care units, beds, ventilators and appropriated therapeutic. Also, human resources were constantly under pressure, at risk to be contaminated, which made them use unusual Personal Protective Equipment, taking longer time to dress and preparation. Similar situations were felt and lived worldwide [12,14,23].

## Limitations and future directions

Data on hospital mortality was not collected but it would give significant information to this investigation. As well, to perform analysis including non-elevation syndromes in addition to those with ST elevation should be considered in future papers.

## Conclusion

During the term of the SFL initiative in Portugal, there were positive changes in indicators of patient delay, such as lower proportions of patients who resorted to primary health centres and local hospitals without PPCI facilities, as well as an increase in the proportion of those who called the EMS number. These results were more significant in early years of SFL implantation, mostly as a result of the public awareness campaign. In the subsequent years this effect was attenuated, probably because the campaign became too repetitive or less intense, which may lead the audience to become bored and stop paying attention, weakening its efficacy. System delay did not significantly change over this period. Moreover, it was even worse due to an unexpected situation, the Covid-19 pandemic. Covid-19 exposed health systems weaknesses, and clearly demonstrate that something unexpected can be a huge draw-back in healthcare programmes. This leads to deterioration of healthcare standards that have been refined over the years These lessons must be taken into account in the future strategies and decision-making, particularly in strengthening current educational programmes aiming to shorten patient and system delays and be prepared for the unexpected.

## Acknowledgments

The authors gratefully acknowledge all Centres Participating in "Stent for Life" Initiative Portugal sponsored by the Portuguese Association of Cardiovascular Intervention—*Associação Portuguesa de Intervenção Cardiovascular* (APIC):

Hospital Vila Real (Henrique Cyrne Carvalho, MD and Paulino Sousa, MD), Hospital Braga (João Costa, MD), Hospital S. João (João Carlos Silva, MD), Hospital Santo António (Henrique Cyrne Carvalho, MD), Centro Hospitalar Vila Nova de Gaia (Vasco Gama Fernandes, MD and Pedro Braga, MD), Hospital de Viseu (João Pipa, MD and Luís Nunes, MD), Centro Hospitalar de Coimbra (Marco Costa, MD and Vitor Matos, MD), Hospital de Leiria (João Morais, MD and Jorge Guardado, MD), Hospital Fernando da Fonseca (Pedro Farto e Abreu, MD), Hospital de Santa Maria (Pedro Canas da Silva, MD and Pedro Cardoso, MD), Hospital Santa Cruz (Manuel Almeida, MD), Hospital de Santa Marta (Rui Ferreira, MD), Hospital Curry Cabral (Luis Morão, MD), Hospital Pulido Valente (Pedro Cardoso, MD), Hospital Garcia de Orta (Hélder Pereira, MD), Hospital Setúbal (Ricardo Santos, MD), Hospital de Évora (Lino Patrício, MD and Renato Fernandes, MD), Hospital de Faro (Victor Brandão, MD and Hugo Vinhas, MD).

## Author Contributions

**Conceptualization:** Ernesto Pereira.

**Data curation:** Ernesto Pereira, Rita Calé.

**Formal analysis:** Ernesto Pereira, Rita Calé, Ângela Maria Pereira.

**Investigation:** Ernesto Pereira.

**Methodology:** Ernesto Pereira, Ângela Maria Pereira.

**Project administration:** Ernesto Pereira.

**Software:** Rita Calé.

**Supervision:** Hélder Pereira, Luís Dias Martins.

**Validation:** Ernesto Pereira, Rita Calé, Ângela Maria Pereira, Hélder Pereira.

**Writing – original draft:** Ernesto Pereira.

**Writing – review & editing:** Ernesto Pereira, Ângela Maria Pereira, Luís Dias Martins.

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
