## [Decision Letter · Decision Letter 0]

5 Jul 2023

PONE-D-23-10215Stent For Life Initiative in Portugal: progress through years and Covid-19 ImpactPLOS ONE

Dear Dr. Pereira,

Thank you for submitting your manuscript to PLOS ONE. After careful consideration, we feel that it has merit but does not fully meet PLOS ONE’s publication criteria as it currently stands. Therefore, we invite you to submit a revised version of the manuscript that addresses the points raised during the review process.

We look forward to receiving your revised manuscript.

Kind regards,

Angélica Baptista Silva, Ph.Sc.

Academic Editor

PLOS ONE

Reviewers' comments:

Reviewer's Responses to Questions

**Comments to the Author**

1. Is the manuscript technically sound, and do the data support the conclusions?

Reviewer #1: Yes

Reviewer #2: Yes

2. Has the statistical analysis been performed appropriately and rigorously? 

Reviewer #1: Yes

Reviewer #2: Yes

3. Have the authors made all data underlying the findings in their manuscript fully available?

Reviewer #1: Yes

Reviewer #2: Yes

4. Is the manuscript presented in an intelligible fashion and written in standard English?

Reviewer #1: Yes

Reviewer #2: Yes

5. Review Comments to the Author

Reviewer #1: The authors sought to describe the effects of an awareness campaign on STEMI in order to improve patient access to primary percutaneous coronary intervention (PPCI). They found that more patients received PPCI after the campaigned was implemented but patient and system delays did not significantly change overtime, probably due to low efficiency of the campaigns in the las years and due to COVID-19 pandemic.

This is an interesting study. PPCI is a life saving therapy for STEMI and all the efforts to increase its use are welcome. The manuscript is well written and the methods and results are clearly demonstrated. In order to improve the manuscript, I suggest including data on hospital mortality as well, in case the authors have collected data on that.

Reviewer #2: Dear Authors,

I congratulate the authors for the excellent paper. It brings applicable information about public policies for the care of patients with acute coronary syndromes. It became clear that campaigns alone are not enough. It is necessary to train care teams, optimize resources and protocols.

The negative influence of the COVID pandemic on cardiovascular mortality and morbidity was consistent with other published papers. It adds the measurement of lack of knowledge about the real severity of acute coronary syndromes by the lay population.

My suggestion for future papers, considering the limitations of the database, to perform analysis of the different clinical presentations of coronary syndromes in addition to those with ST elevation. We know that non-elevation syndromes predominate.

6. PLOS authors have the option to publish the peer review history of their article (what does this mean?). If published, this will include your full peer review and any attached files.

Reviewer #1: No

Reviewer #2: **Yes: **Wolney de Andrade Martins

---

## [Author Response · Author response to Decision Letter 0]

13 Aug 2023

Concerning Journal Requirements:

Revised and corrected.

2. Please provide additional details regarding participant consent. In the ethics statement in the Methods and online submission information, please ensure that you have specified what type you obtained (for instance, written or verbal, and if verbal, how it was documented and witnessed).

Information has been completed and detailed 

3. We note that you have stated that you will provide repository information for your data at acceptance. Should your manuscript be accepted for publication, we will hold it until you provide the relevant accession numbers or DOIs necessary to access your data. If you wish to make changes to your Data Availability statement, please describe these changes in your cover letter and we will update your Data Availability statement to reflect the information you provide

Although we stated we would make the Data fully available without restriction, they were gathered from a national database, from the Portuguese Society of Cardiology, and we are not authorized to provide them for a repository. We were not aware of this at the moment of the submission, however, we believe in open science and therefore we would like to change our Data Availability statement to the option that allows sharing the data sets available only upon request and we will make efforts to assure access.

Reference list was reviewed and no changes were made.

Response to Reviewers 

We thank the kind comments from the reviewers, which made us proud of our work.

Reviewer 1 made an important comment in order to improve the manuscript, suggesting “including data on hospital mortality as well, in case the authors have collected data on that.” We totally agree with this comment, but as matter of fact these data were not collected. Nevertheless we included this suggestion in a new section named Limitations and Future directions.

Reviewer 2 made also an important comment in order to be included in future papers suggesting “performing analysis of the different clinical presentations of coronary syndromes in addition to those with ST elevation”. This suggestion was also included in the new section named Limitations and Future directions.

---

## [Editor Report · Acceptance letter]

17 Oct 2023

PONE-D-23-10215R1 

Stent For Life Initiative in Portugal: progress through years and Covid-19 impact 

Dear Dr. Pereira:

I'm pleased to inform you that your manuscript has been deemed suitable for publication in PLOS ONE. Congratulations! Your manuscript is now with our production department. 

Kind regards, 

on behalf of

Dr. Chiara Lazzeri 

Academic Editor

PLOS ONE